# Understanding the Asymptotic Performance of Model-Based RL Methods

## Abstract

In complex simulated environments, model-based reinforcement learning methods typically lag the asymptotic performance of model-free approaches. This paper uses two MuJoCo environments to understand this gap through a series of ablation experiments designed to separate the contributions of the dynamics model and planner. These reveal the importance of long planning horizons, beyond those typically used. A dynamics model that directly predicts distant states, based on current state and a long sequence of actions, is introduced. This avoids the need for many recursions during long-range planning, and thus is able to yield more accurate state estimates. These accurate predictions allow us to uncover the relationship between model accuracy and performance, and translate to higher task reward that matches or exceeds current state-of-the-art model-free approaches.

## 1 Introduction

Model-based reinforcement learning (MBRL) has many potential benefits over model-free approaches. These include (i) the ability to generalize to new tasks in the environment, without having to retrain; (ii) learning from off-policy data and (iii) sample efficiency. However, in simulated environments where data is plentiful, model-based approaches struggle to approach the asymptotic performance of model-free methods Nagabandi et al. (2017); Pong et al. (2018); Chua et al. (2018). Several possible explanations present themselves: the planner used for selecting optimal actions under the model might be insufficiently powerful; the model might not be able to accurately model the dynamics; or the planning horizon might not be long enough.

This paper address these questions by teasing apart the different factors involved in an MBRL framework, applied to two deterministic MuJoCo environments (Todorov et al., 2012), with the aim of understanding the gap in asymptotic performance with respect to model-free approaches. In particular, we demonstrate that bias caused by short planning horizons and poor accuracy of long-term predictions is the cause of the poor performance of existing MBRL methods in the unlimited-sample regime. Our experiments show that, with a perfect dynamics model, the optimal planing horizon can be over 100 steps – much longer than typically considered in many MBRL approaches. Correspondingly, the performance is typically limited by the ability of the model to accurately predict over long-time scales, not just a few time-steps.

Existing approaches to MBRL rely on a single-step dynamics model that predict the next state, given the current state and an action. As can be see in Figure 5, over long time-horizons the errors compound due to recursive application of the model, yielding inaccurate state estimates which are not useful for planning. Instead, we propose an alternate form of dynamics model that takes as input a *sequence* of actions along with the current state and directly predicts many time-steps into the future. This approach provides accurate prediction over long time horizons, allowing us to uncover the relationship between model accuracy and performance. This reveals that MBRL with sufficiently good learned models matches or exceeds the performance of state-of-the-art model-free methods.

### 1.1 Related Work

**Non-Parametric Model-Based RL:** Gaussian processes are popular approach to modeling non-linear dynamics due to their low sample complexity and their ability to explicitly represent epistemic uncertainty. Consequently, numerous MBRL approaches use them, e.g. Kocijan et al. (2004); Ko

et al. (2007); Grancharova et al. (2008); Deisenroth & Rasmussen (2011); Deisenroth et al. (2014). However via the choice of kernel they impose potentially unrealistic smoothness constraints and do not scale to large data settings, limiting their asymptotic performance in practice.

**Combining model-based and model-free methods:** Due to the sample efficiency of model-based methods and superior asymptotic performance of model-free methods, several works have proposed to learn dynamics models using a few trajectory samples, then use those models to train or augment a model-free policy. The classic Dyna algorithm (Sutton, 1990) uses a model to extend Bellman updates multiple steps. Deisenroth & Rasmussen (2011) learns a Gaussian process model of the dynamics function and uses it to train an RBF network policy, and Gal et al. (2016) enables the model to scale to larger data by using Bayesian neural networks in place of GPs. Levine et al. (2016) fits a time-varying locally linear model around a trajectory, then trains a neural network policy to follow trajectories found by iLQR (Todorov & Li, 2005). Silver et al. (2016) learns an implicit model of the dynamics for implicit planning via value estimation; in an inversion of this technique, Pong et al. (2018) learn an explicit model of Q values for explicit planning via constrained optimization. Weber et al. (2017) learns a neural network dynamics model which is unrolled inside a policy to inform an actor-critic agent. Nagabandi et al. (2017) trains a neural network dynamics model on control tasks and uses it to actions, then uses that model-based policy to speed the training of a model-free policy via imitation learning. These works largely seek to either (i) augment a model-free method with a model for faster learning, or (ii) make up for the asymptotic deficiencies of a model-based method by transitioning to model-free. In this work we instead directly investigate the causes of MBRL's poor asymptotic performance with the aim of making a transition to model-free unnecessary.

**MBRL with neural network models:** The idea of using neural networks to enable model-based control of nonlinear systems goes back decades (Miller et al., 1990; Schmidhuber, 1990; Hunt et al., 1992; Bekey & Goldberg, 2012; Draeger et al., 1995), but until recently has only seen significant success on systems with relatively simple dynamics. Several works have endeavored to use neural network generative models of images for model-based control (Wahlström et al., 2015; Watter et al., 2015; Finn & Levine, 2017); these policies have typically used short planning horizons and struggled to equal model-free performance on complex tasks. Lenz et al. (2015) learn recurrent neural network dynamics models, then use backpropagation through time to select actions and control a robotic arm, and Henaff et al. (2017) extend this concept to both discrete and continuous action spaces. Clavera et al. (2018) combine meta-learning with MBRL using neural network models to rapidly adapt to novel environments. Srinivas et al. (2018) uses imitation learning to train a model to plan by gradient descent, which relies on an existing expert rather than learning from scratch.

The closest work to ours is Chua et al. (2018). This follows a similar recipe, with similar planning and constructing a dataset online, but with different models and different goals. We use deterministic neural networks which predict many steps into the future to understand the impact of model- and horizon-bias on asymptotic performance of MBRL methods on long-horizon problems. Chua et al. (2018) uses a bootstrapped ensemble of probabilistic neural networks to improve the performance of MBRL in the few-sample regime. While that work achieves strong performance on short-horizon tasks, in our experiments we find it struggles to equal model-free methods on tasks with very long horizons.

## 2 APPROACH

In this section we describe the models used in our experiments, the action-conditional predictor (ACP) and the novel plan-conditional predictor (PCP), which predicts the outcome of a sequence of actions with a single model step. We then detail the framework we use for planning with and training these models.

### 2.1 NOTATION

We denote states and actions at a time $t$ by $s_t$ and $a_t$. In the environments we consider, both $s_t$ and $a_t$ are continuous vectors. We use $H$ to refer to the planning horizon of an MPC policy. We refer to a sequence of actions as a *plan*; a plan constructed with horizon $H$ is thus $p = \{a_1, ..., a_H\}$. In a set of plans $\{p^1, ..., p^n\}$, $a_j^i$ refers to the $j$th action of the $i$th plan.

We consider models which predict a future state given the current state and one or more actions. We denote $R$ to be the *range* of such a model, which is the number of steps this model predicts in a single application: $f_R(s_t, a_t, \ldots, a_{t+R-1}; \theta) = \tilde{s}_{t+R}$, for parameters $\theta$.

We apply the model recursively using the notation $F_T(s_t, a_t, \ldots, a_{T-1}; \theta) = f_R(\ldots f_R(f_R(s_t, a_t, \ldots, a_{t+R-1}; \theta), a_{t+R}, \ldots, a_{t+2R-1}; \theta) \ldots, a_{T-R}, \ldots, a_{T-1}; \theta) = \tilde{s}_T$. That is, $F_T()$ applies $f_R()$ recursively $T/R$ times.

## 2.2 PLAN-CONDITIONAL PREDICTORS

To test our conjecture that compounding errors limit the asymptotic performance of existing models on long-horizon RL tasks, we propose plan-conditional predictors (PCPs). A PCP takes the form $f_R(s_t, a_t, \ldots, a_{t+R-1}) = \tilde{s}_{t+R}$ for some range $R > 1$. If $R = 1$, then it reduces to the standard approach (Deisenroth & Rasmussen, 2011; Gal et al., 2016; Henaff et al., 2017; Nagabandi et al., 2017; Chua et al., 2018) which predicts only a single time-step at a time, and which we call an action-conditional predictor (ACP). As shown in Figure 1, a PCP can predict $H$ steps into the future using $H/R$ recursive applications of the model instead of $H$ applications required by an action-conditional predictor.

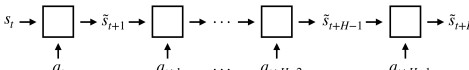 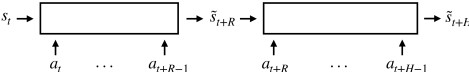

(a) Action-conditional predictors must recurse $H$ times to make a prediction $H$ steps into the future.

(b) A plan-conditional predictor with range $R$ only needs to recurse $H/R$ times (shown here as twice).

Figure 1: Action-conditional vs. plan-conditional predictors.

There are many possible parameterizations that could be used in a PCP. In this work we choose deep fully-connected neural networks. There are several reasons for this: (i) the space of inputs grows exponentially with the range $R$, thus models with high capacity are needed to minimize model bias; (ii) since the goal is to understand asymptotic performance, a large data regime is assumed and sample complexity is a secondary issue; (iii) by predicting R steps with a single network application, they are extremely fast at planning time. An obvious alternate parameterization is recurrent neural networks (RNNs), and we provide experimental comparisons between the two approaches in Section 3.

For both `Swimmer` and `HalfCheetah`, the ACP and PCP networks consist of fully-connected networks with 9 hidden layers with 1000 units each using the SELU activation function (Klambauer et al., 2017). The input state and action(s) are concatenated before being used as input to the network.

The same loss function is used for training PCPs and ACPs, namely $\|\tilde{s}_{t+R} - s_{t+R}\|_2^2$, where $s$ is the raw MuJoCo state. We assume that the environment forms a Markov decision process (MDP) (Bellman, 1957) with deterministic dynamics, properties shared by the tasks considered in this work. These assumptions allow us to focus exclusively on the significance of model fidelity and planning horizon to MBRL, but removing them is an interesting direction for future work.

### 2.2.1 INTERMEDIATE PREDICTIONS

For visualization purposes, in some experiments we use a variant of a plan-conditional predictor which takes as input a state and a variable number of actions $R'$ where $1 \leq R' \leq R$, the remainder of action inputs being set to zero. This allows us to plot error or render video of the PCP's predictions at each intermediate timestep instead of only at multiples of $R$.

Furthermore, this variant of the model allows for planning based on reward functions which operate at each timestep rather than just at the end of the episode (see Section 2.3.1). As such, this variant of the model is applicable to any environment.

## 2.3 SELECTING OPTIMAL ACTIONS

In order to turn a predictor into a policy, we employ an off-the-shelf planning approach, namely the cross-entropy method (CEM) (Botev, 2011), to find a plan that is optimal up to some horizon $H$. We

take the first action from that plan and then replan, a technique known as model-predictive control (MPC) or receding-horizon control (Mayne & Michalska, 1990).

### 2.3.1 Planning with Cross-Entropy Method

Given a predictor $F$, a horizon $H$, and a reward function $r$, we would like to find an optimal plan:

$$p_t^* = \underset{a_t, \ldots, a_{t+H-1}}{\arg\max} \sum_{i=0}^{H-1} r(\tilde{s}_{t+i}, a_{t+i}) \mid \tilde{s}_{t+i} = F_H(s_t, a_t, \ldots, a_{t+i-1})$$

For both MuJoCo environments, the reward function is dominated by the distance traveled in the $x$-dimension at each timestep[1]. Thus for planning purposes we can replace the original reward function with a sparse one which provides reward equal to the $x$-distance traveled at the end of the episode: $\hat{r}(s_t) = \begin{cases} s_t[x] & \text{if } t == T \\ 0 & \text{otherwise} \end{cases}$. We then substitute $t + H$ for $T$ and plan based on $\hat{r}(s_{t+H})$; note this is identical to using the sum of $x$-progress at each timestep $t...t + H$. This reduces the form of the optimal plan under a predictor $F$ to

$$p_t^* = \underset{a_t, \ldots, a_{t+H-1}}{\arg\max} \ \hat{r}(F_H(s_t, a_t, \ldots, a_{t+H-1}))$$

The cross-entropy method starts with a set of plans $p$ drawn from a candidate distribution $C$. In continuous control tasks, sampling actions independently along a trajectory results in near-zero net motion. Therefore it is common to instead use correlated action noise for exploration or trajectory sampling, e.g. an Ornstein-Uhlenbeck process (Uhlenbeck & Ornstein, 1930) as in DDPG (Lillicrap et al., 2015). We define the candidate $C(\cdot)$ distribution by the following sampling process:

$$a_0 \sim \mathcal{U}(-1, 1)$$
$$a_{t+1} = \min(\max(\mathcal{N}(\mu = a_t, \sigma = 0.2), -1), 1)$$

i.e. the sampled actions are clamped to be in the range $\pm 1$ which are the limits of the action space.

The overall planning framework is shown in Figure 2. After drawing an initial set of $N$ plans $\{p^1, \ldots, p^N\} \sim C(\cdot)$, these are passed through the predictor $F$ to estimate rewards $r^1, \ldots, r^N$, which are used to rank the plans. The top $K$ are then passed to a 2nd round (red box). Additionally, their mean and variance are computed[2] and used as parameters for a Gaussian distribution from which $N - K$ new plans are sampled (green box). The combined set of plans are then passed to the PCP to rank them. The output from the planner is the first action from the top-ranked plan (yellow box) at the final planning round, which is executed by the agent in the environment.

The top $K$ trajectories from the final round are used to seed the initial set of plans for replanning at the next timestep, after clipping the first action from each. In practice, we use 3 rounds of planning at each timestep, i.e. two rounds of resampling (the figure omits the 3rd round for clarity). In our experiments, we use $N = 50$ and $K = 5$.

---

[1]In all experiments we *evaluate* a policy using the original reward function from OpenAI Gym. This simplified reward function is exclusively used inside the policy. We found in experiments using the ground-truth dynamics as a model that planning with the true reward function instead made no significant difference on these tasks.

[2]Independently at each timestep and action dimension; $\mu(p^1, \ldots, p^n) = \{\frac{1}{n}\sum_i a_j^i \mid j \in H\}$ and $\sigma(p^1, \ldots, p^n) = \left\{\sqrt{\frac{1}{n}\sum_i \left(a_j^i - \mu(a_j^{1 \cdots n})\right)^2} \mid j \in H\right\}$.

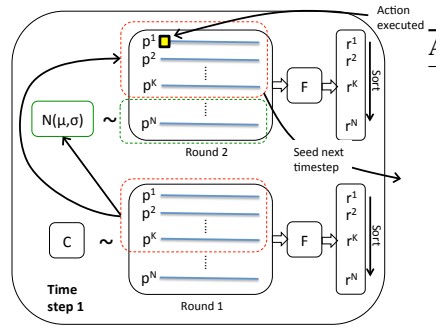

Figure 2: Our off-the-shelf planner, based on the cross-entropy method (Botev, 2011). See text for details.

---

**Algorithm 1** On-policy data aggregation and training

---

Initialize dataset $\mathbb{D}$ with trajectories from random policy
**while** not converged **do**
$\quad \theta \leftarrow \arg\min_{\Theta} \mathbb{E}_{s_t, a_{t...t+R-1}, s_{t+R} \sim \mathbb{D}} \left(f_R(s_t, a_{t...t+R-1}) - s_{t+R}\right)^2$
$\quad$ **for** $m = 0...M$ **do**
$\quad\quad$ **for** $t = 0...T$ **do**
$\quad\quad\quad s_t \leftarrow$ Env.get_observation()
$\quad\quad\quad p_t \leftarrow \arg\max_{a_{t...t+H-1}} \hat{r}(F_H(s_t, a_{t...t+H-1}; \theta))$
$\quad\quad\quad a_t \leftarrow$ head$(p_t)$
$\quad\quad\quad$ Env.execute$(a_t)$.
$\quad\quad$ **end for**
$\quad\quad \mathbb{D} \leftarrow \mathbb{D} \bigcup (s_{0...T}, a_{0...T-1})$
$\quad$ **end for**
**end while**

---

### 2.4 ONLINE TRAINING

The planning framework described above turns the PCP model into a policy which outputs an action at each time step. To train this policy, the underlying PCP model must be updated in an online fashion.

This requires a dataset of trajectories $\{s_{0...T}, a_{0...T-1}\}$ that covers the environment's state-action space. We follow Nagabandi et al. (2017); Chua et al. (2018) and others and collect this dataset by alternating between fitting the model to the existing data and using our planning procedure (2.3.1) to generate more trajectories from the environment. We collect trajectories from the environment for $M = 100$ episodes. These trajectories are added to the training set and the PCP model is updated with SGD for 10 epochs using AMSGrad (Reddi et al., 2018). The overall procedure is detailed in Algorithm 1 and is essentially the standard template for MBRL (Deisenroth & Rasmussen, 2011; Gal et al., 2016; Nagabandi et al., 2017; Chua et al., 2018).

## 3 EXPERIMENTS

### 3.1 EFFECT OF PLANNING HORIZON AND MODEL RANGE

In this experiment we directly test our hypothesis that plan-conditional predictors are able to benefit from longer planning horizons than action-conditional predictors. Figure 3 shows that while an ACP model is competitive for planning horizons up to 20 timesteps, its performance falls substantially below the PCP models by 40 timesteps. The ACP model scores best at a horizon of 60 timesteps; beyond that its performance degrades as its predictions become unusably inaccurate.

By contrast the PCP models shown, which need only be recursively applied between 3 and 20 times, all show monotonically increasing rewards as the planning horizon is increased. This reveals two things: the `Swimmer` task has a minimum optimal planning horizon of at least 100 timesteps, and the PCP models are able to predict with sufficient accuracy to be useful even over that long horizon.

In the next experiments we tease apart the different factors of planning horizon, range, and accuracy that combine to produce these results.

### 3.2 PLANNING HORIZON VS REWARD

Using a ground-truth model of the environment (that is, MuJoCo itself) in conjunction with our planner we can look at performance as a function of planning horizon for these tasks. That is, we define a new predictor $\text{MJC}(s_t, a_{t...t+T})$ which can make predictions arbitrarily far into the future. This predictor works internally by creating a new copy `env_internal` of the Gym environment. To make a prediction $\text{MJC}(s_t, a_{t...t+T}) = s_{t+1}$, this predictor will call `env_internal.set_state(s_t)`, then repeatedly call `env_internal.step(a_{t+i})` for $i = 0...T$. Its output is the observation after the final action has been executed. This ground-truth predictor can be used for planning like any other.

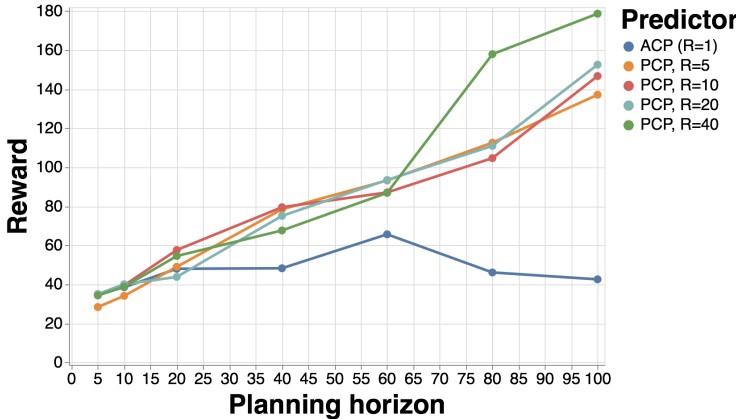

Figure 3: Reward attained at convergence on `Swimmer` by action-conditional predictors (Range = 1) and plan-conditional predictors (Range > 1) trained using different planning horizons. For this experiment we use the PCP variant which can make intermediate predictions to allow planning horizons that are not even multiples of ranges, e.g. a range 40 model with horizon 60.

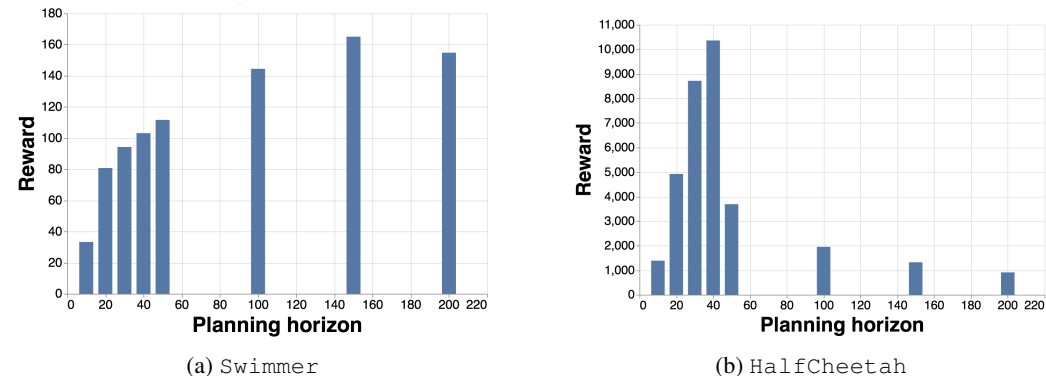

(a) `Swimmer`                              (b) `HalfCheetah`

Figure 4: Reward as a function of planning horizon for `Swimmer` and `HalfCheetah` using ground-truth dynamics with our planner. `Swimmer` performance increases up to a horizon of $H = 150$. For `HalfCheetah` the optimal horizon is around $H = 40$, after which performance decreases due to the instability of the simulation when using single- instead of double-precision floating point.[3]

The results of this experiment, shown in Figure 4, show that the optimal planning horizons for `Swimmer` and `HalfCheetah` are at around 150 and 40 timesteps, respectively. These results provide additional clarity to those presented in the previous experiment. Policies that use the ground-truth dynamics as their predictor perform better as the horizon increases due to the decreased bias of the long-horizon reward estimate. The approximate dynamics model from PCPs show similar gains. However, beyond a certain planning horizon the quasi-random search in the planner becomes less effective due to variance caused by the huge size of the search space, causing the reward to dip for $H > 150$.

Previous work (Nagabandi et al., 2017; Chua et al., 2018) has demonstrated planning for 20 or 30 timesteps with a neural network dynamics model (and in particular, the recent Chua et al. (2018) achieves impressive scores on HalfCheetah). However, to our knowledge planning horizons above 50 timesteps remain untested. This leads us to investigate the accuracy at very long range prediction of traditional action-conditional predictors as well as our plan-conditional predictors.

---

[3]We found it necessary to handicap the ground-truth model on `HalfCheetah` by adding noise to its actions during planning to prevent the planner from breaking the simulation. Without this handicap the planner was able to find nonphysical strategies and achieves expected reward of up to 150,000.

### 3.3 MODEL RANGE VS ACCURACY

With the evidence from Section 3.2 that some tasks require planning horizons up to 150 timesteps, we now evaluate action-conditional and plan-conditional predictors on their ability to make long-range predictions in MuJoCo. Additionally we compare to an RNN which predicts one step at a time, but which is trained with backpropagation through time (BPTT) to minimize prediction error across all timesteps. This RNN is trained with a curriculum of prediction lengths ranging from 1 (at the beginning of training) to 200 (at the end).

To enable direct comparisons, we employ a fixed dataset of trajectories from the environment. We generate this dataset by training a model-free PPO (Schulman et al., 2017) agent on `Swimmer` and recording the trajectories that it takes. This ensures that the dataset contains trajectories that involve interacting with the environment in nontrivial ways. We then split that data into a training set and a validation set and train each model to convergence on the training set.

Figure 5 shows the results of evaluating these models on the validation set. While the ACP is able to make extremely accurate predictions a few steps into the future, it suffers from accumulating error when it is recursively applied many times. This suggests an explanation for the inability of the ACP models to take advantage of planning horizons longer than 60 timesteps, as discussed in Section 3.1. As the horizon increases the predictions from an ACP diverge from reality, while simultaneously the bias from using a too-short planning horizon decreases. This produces an optimal planning horizon of intermediate length given a model with error that increases as a function of depth.

The RNN and the PCP are both optimized for long-term prediction accuracy and thus make much better predictions. The PCP model achieves slightly better accuracy, and does it in a fraction of the time; to predict 200 steps in the future requires 200 applications of the RNN network, but only 4 of the PCP. This performance gap is significant, as planning requires evaluating hundreds of thousands of trajectories per episode.

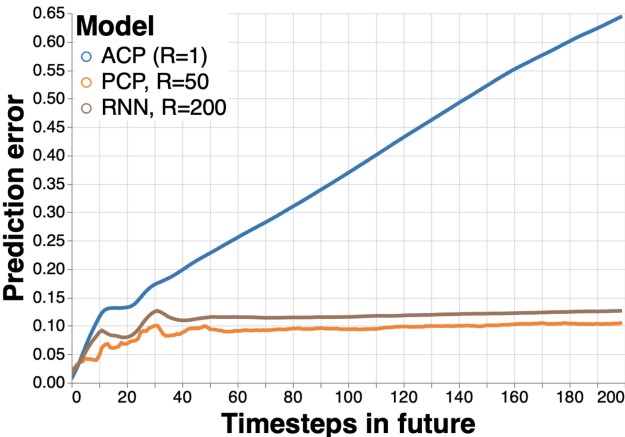

Figure 5: Mujoco `Swimmer` task. Prediction error in estimated distance traveled as a function of timesteps in the future for two models trained on the same dataset of off-line trajectories: a 1-step predictor (blue), an RNN trained with 200 steps of BPTT (brown), and a 50-step predictor (orange). The 1-step model, which is standard in the literature, exhibits poor long-term predictions due to accumulating accumulating errors. By contrast, the 50-step model and the RNN are trained to minimize long-term error.

### 3.4 MODEL ACCURACY VS REWARD

Figure 6 shows the reward vs prediction error for 10 models at various points during online training in the `Swimmer` environment. These results show a clear relationship between low prediction error of the model and high reward, reinforce the importance of having a highly accurate long-range model of the environment. Taken together with the high error for ACPs in Figure 5, this explains that the RL performance of action-conditional predictors is limited by their inability to make accurate predictions at long timescales.

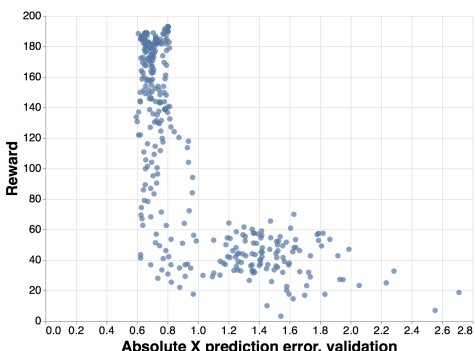 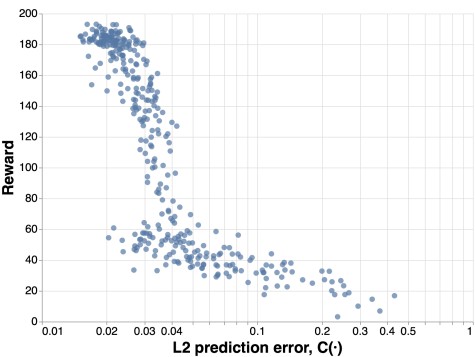

Figure 6: Experiments on `Swimmer` showing the relationship between the predictor's accuracy at $H = 100$ steps into the future and the reward obtained by using that predictor for MPC with a horizon of $H$ timesteps. These plots include five seeds each of ACP and PCP ($R = 50$), with each point representing a particular predictor at some point in the training process. **Left**: Absolute error at predicting the X-coordinate $H$ steps in the future along a trajectory taken by a trained PPO agent. Our policies select actions based on which plan results in the greatest X-coordinate prediction. **Right**: $\ell_2$ loss at predicting the outcome from following a plan sampled from the candidate distribution $C$.

The left panel of Figure 6 shows a very vertical trend at the far left of the plot. We hypothesize that this is due to the changing distribution of the training data as a function of the predictor's accuracy; once a predictor is making accurate predictions, the trajectories that it follows change from being nearly random to more focused. This then means that most of the progress late in training is on refining the predictions on a very narrow distribution of trajectories. These refinements continue to improve the RL performance of the predictor but have little impact on its accuracy along trajectories coming from a different policy.

## 3.5 COMPARISON TO OTHER APPROACHES

In this experiment we evaluate the performance of ACP and PCP models compared to previous reinforcement learning methods, both model-free and model-based, on `Swimmer-v2` and `HalfCheetah-v2` from OpenAI Gym (Brockman et al., 2016)[4]. We also show the performance of an RNN model, which is identical to ACP but trained via backpropagation through time (BPTT) to minimize prediction error across the entire planning horizon, i.e. 100 steps for `Swimmer` and 20 steps for `HalfCheetah`. Since the PCP models predict $R$ timesteps per network application (versus one forward pass through the network per timestep for ACP and RNN models) the PCP models are a factor of $R$ faster to plan with in wall-clock time.

Our main model-based baseline is PETS (Chua et al., 2018), a state of the art probabilistic neural network-based MBRL algorithm which has been shown to equal or exceed model-free performance on short-horizon tasks. Similar to this work, PETS uses MPC and CEM for model-based control and aggregates a dataset online. On `HalfCheetah` we also compare with the model-based results from Nagabandi et al. (2017), which follows the same basic formula as our work but uses random shooting to find optimal plans instead of CEM. Our model-free baseline is PPO (Schulman et al., 2017) as implemented by Kostrikov (2018), a high-performing actor-critic method.

For each method we run five seeds and allow the algorithm to run to convergence, as we are interested in evaluating asymptotic performance. We train PPO for 100,000 trajectories. On `Swimmer` and `HalfCheetah` we use planning horizons of 100 and 20 timesteps respectively for the ACP, PCP, RNN, and PETS results. For each of the baseline methods we plot a horizontal line indicating the best score achieved by that method at any point in training, after averaging over the random seeds and over several consecutive episodes. We also show lines for the score achieved by using the ground-truth dynamics with the same planner as we use for PCP. In the case of Nagabandi et al. (2017) the score shown is that reported in their work; while the version of the `HalfCheetah` environment that they use is slightly different from the Gym one, we believe the numbers to be roughly comparable.

---

[4]We selected `Swimmer` and `HalfCheetah` to follow the main experiments from Nagabandi et al. (2017).

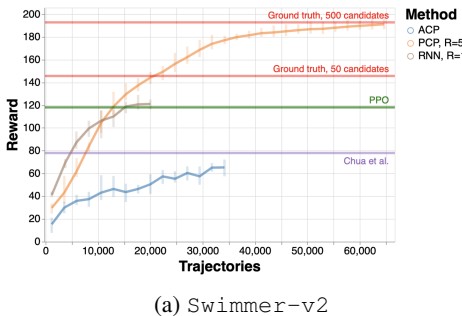 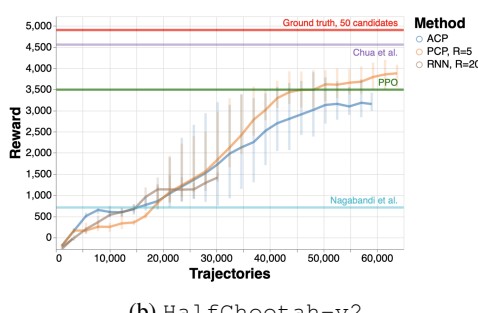

(a) `Swimmer-v2`  (b) `HalfCheetah-v2`

Figure 7: These plots show the scores achieved by ACP, PCP, RNN, and methods previously published in the literature. Solid lines indicate average reward; translucent bars indicate the best and the worst performance across 5 seeds. **Left**: On `Swimmer` PCP achieves rewards of 185, outperforming any previous model-based or model-free method. In particular, the next-best model-based approach reaches scores less than half those attained by PCP. RNN models outperform ACP but lag behind PCP. **Right**: On `HalfCheetah` PCP equals the rewards attained by PPO. PCP outperforms ACP and Nagabandi et al. (2017) despite the planning horizon being a short $H = 20$.

Figure 7 shows the results of this experiment. On `Swimmer`, which has a very long planning horizon, PCP achieves rewards more than 50% higher than the next-best method, while on `HalfCheetah` it equals the performance of PPO. We speculate that the extremely high rewards achieved by Chua et al. (2018) on `HalfCheetah` are due in part to the difference in settings for CEM betweeen our two works; while we use 3 steps of CEM optimization with 50 candidates per step, Chua et al. (2018) use 5 steps of optimization on 500 candidates.

## 4  DISCUSSION

In this work we considered the problem of model bias in model-based reinforcement learning. In the largely deterministic environments considered, we show that optimal planning horizons can be large, beyond 100 timesteps. Over these horizons, NN-based models trained to minimize single-step prediction accuracy do not perform well. We demonstrate that better performance is possible with NN models by changing the loss function and the form of the model. Further experiments confirm that model accuracy is crucial to end task performance.

Our experiments make several simplifying assumptions, most notably the availability of unlimited samples and deterministic environment dynamics. Sample complexity would undoubtedly be improved by replacing the current overparameterized MLP architecture with something more efficient, an interesting future direction. Another important area for future work is understanding the interaction of long-range planning with stochasticity in the environment, including the development of generative models capable of predictions over long horizons.

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

## APPENDIX A    PLANNING WITH A GROUND TRUTH MODEL

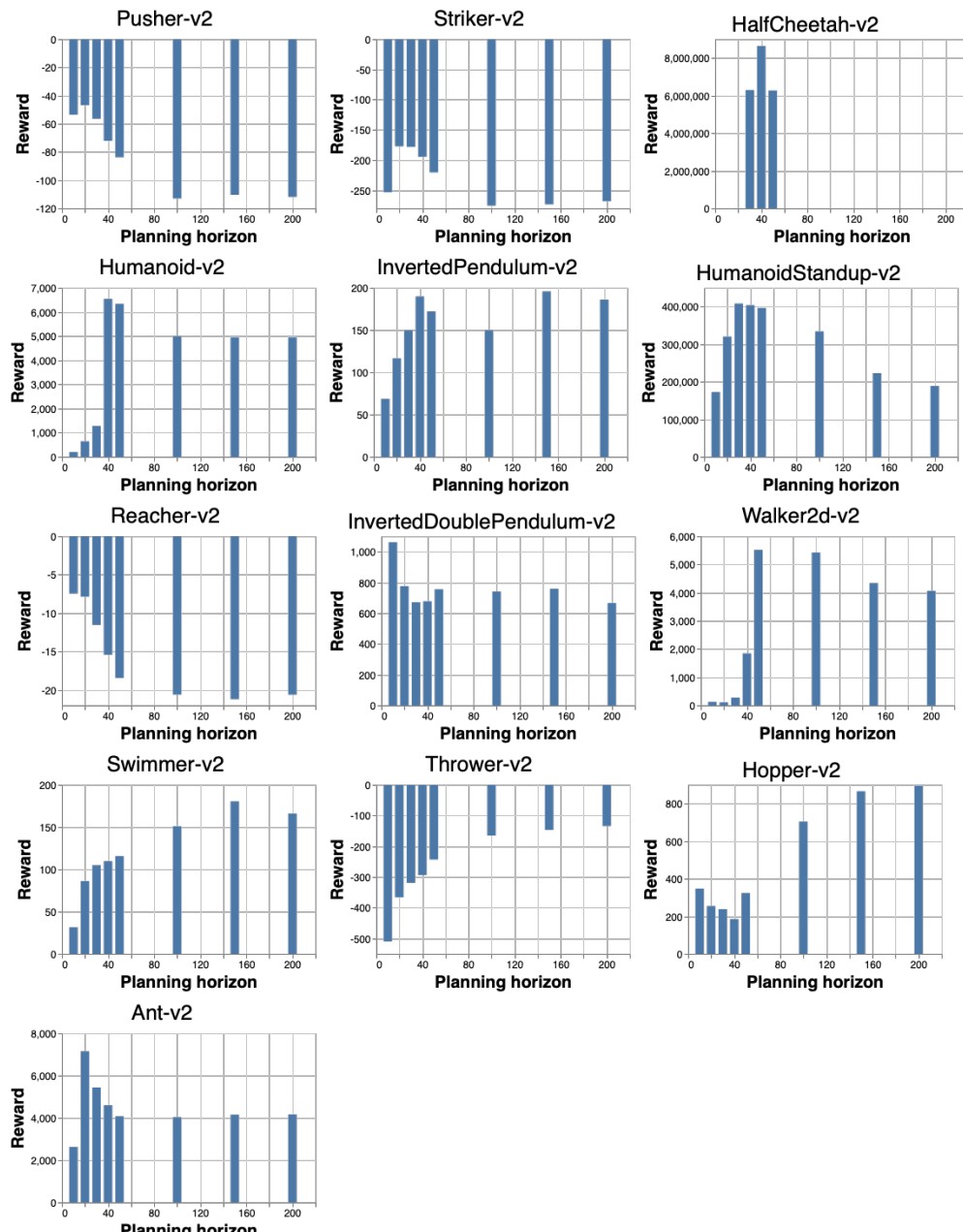

Figure 8: Results from using our CEM optimizer and the ground-truth dynamics to plan with varying horizons across all 13 MuJoCo environments in OpenAI Gym. We believe these results could be helpful for comparing the relative impact of delayed reward in these environments. In all cases CEM uses a pool of 50 candidates and 5 steps of optimization per timestep. Of particular note: (1) In some environments, planning with long horizons performs worse than planning with short horizons. This is largely due to instability in those simulations, resulting in incorrect model predictions when using single- vs double-precision state encodings and long time horizons. (2) In the Cheetah environment, CEM is able to break the simulation and earn rewards greater than should be physically possible.

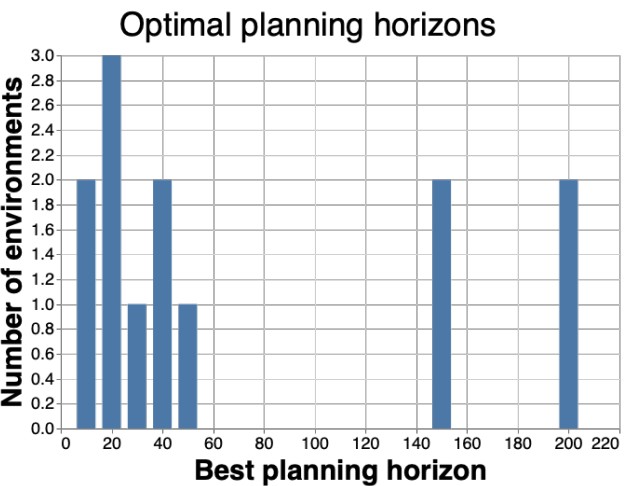

Figure 9: This histogram aggregates the results shown in Figure 8. Of the 13 MuJoCo environments in OpenAI Gym, 4 have optimal planning horizons above 100 timesteps.

