# OpenReview forum: "Understanding the Asymptotic Performance of Model-Based RL Methods"
_ICLR.cc/2019/Conference_

### Official Review · AnonReviewer3 · 2018-11-02
**weak contribution**

**Rating:** 2
**Confidence:** 4

**Review:**

This paper study the model-based approach in deterministic low dimensional continuous control. As far as I am concerned and I understood, the main contribution of this paper is in substituting one-step-ahead prediction model with a multiple-step prediction model, resulting in a more accurate prediction model. I was not able to find points beyond this. I would be happy if the authors could clarify it.

---

> ### Author Response · Authors · 2018-11-26
> **See top-level comment**
>
> Please refer to our top-level comment.

---

### Official Review · AnonReviewer2 · 2018-11-02
**Model-based RL with action sequences, need more convincing regarding applicability**

**Rating:** 4
**Confidence:** 3

**Review:**

This paper proposes learning a transition model that takes an action sequence as an input (instead of a single action), and performing model-based planning by using the cross-entropy method.

One obvious concern with this is that this produces a sequence of open-loop plans, rather than a closed-loop policies, with all the inherent limitations. I could see this working well in practice in problems where anticipating how future decisions will react to state changes is not that important, however the authors should discuss the trade-offs more.

A larger concern for me revolves around learning the transition model. Taking the action sequence as an input (which is one of the main novelties in the paper) is likely to require a lot of data, and maybe this is fine on relatively simple Mujoco tasks but I see it as a potential issue when trying to expand this to more realistic problems.

Finally, I suggest that the authors change the title to something more descriptive of the paper’s contents, as there is no analysis of asymptotic performance in the paper (as I would have thought from the title). I also recommend that they look to see if there is any model-based work in the semi-MDP literature, which could be relevant here.

---

> ### Author Response · Authors · 2018-11-26
> **A few misunderstandings**
>
> “produces a sequence of open-loop plans”: We feel the reviewer may misunderstand how a model-predictive (MPC) controller operates. By replanning at each step, MPC converts an open-loop planner into a closed-loop controller. Please refer to Section 2.3 and Algorithm 1 of our paper for a more precise description, or see https://ieeexplore.ieee.org/document/57020, for example, for more background.
>
> “likely to require a lot of data”: While we agree that our approach does require a significant amount of training data, the main objective of this work is to understand the limitations of existing neural dynamics models rather than to propose a practical new one.
>
> “change the title”: Correspondingly, our paper examines performance in the limit of a very large number of training examples, hence the “asymptotic” in our title.
>
> Thank you. Yes, we will certainly look for related work in the semi-MDP literature.

---

### Official Review · AnonReviewer1 · 2018-11-08
**Incremental advance on model-based reinforcement learning methods**

**Rating:** 6
**Confidence:** 4

**Review:**

The authors learn a model that predicts the state R steps in the future, given the current state and intervening actions, instead of the predicting the next time step state. The model is then used for standard model predictive control. The authors find numerically that their method, termed Plan-Conditional Predictor (PCP), performs better over long horizon times (~100 time steps), than other recent model-based and model-free algorithms. This because for long horizon time scales, the model predicting the state for the next time step accumulates error when used recursively.

The key idea is to use a model that directly predicts multiple time steps into the future. While seemingly an obvious extension, it does not appear to have been used in current algorithms. A main issue that I find with this approach is: since only the state after R steps is predicted, reward r(s_t,a_t) can only be used every R steps, not at every step. The authors gloss over this issue because for both MuJoCo environments that they tested, they only need to consider reward at the end of the planning horizon. Thus to make their algorithm generally applicable, the authors also need to show how or whether their method can deal with rewards that may appear at any time step.

Further, rather than speculate on the cause of the difference between their PCP and PETS (Chua et al 2018) on half-cheetah to be their different settings for CEM optimization (Fig 7b), the authors should just use the same settings to compare. Possibly the authors ran out of time to do this for the current submission, but should certainly do it for the final version.

While the authors have already compared to other algorithms with similar aims, eg Chua et al 2018, they may also wish to compare to a recent preprint Clavera et al Sep 2018, which also aims to combine the sample efficiency of model-based methods while achieving the performance of model-free ones, by using an ensemble of models, over a 200 time step horizon. However, given the recency of this algorithm, I don't consider this essential.

Overall, I feel that the authors idea of an R-step model is worth spreading in the community, if the above two main points are addressed. At the same time, I can only rate it at the border of the cutoff mark.

---

> ### Author Response · Authors · 2018-11-26
> **Some clarifications**
>
> “reward r(s_t, a_t) can only be used every R steps, not at every step”: In fact, our model is able to use reward at intermediate time steps. To do this, we can use the variable-action-length version of our model, as used for visualization (mentioned in Section 2.2 paragraph 2). We also use this version in Fig 5. to evaluate prediction error as a function of timesteps in the future. We apologize for not making this point more clear in the paper and we have revised the paper accordingly, adding a Section 2.2.1.
>
> Comparison btw. PCP & PETS: We directly compared PCP and PETS using the same CEM settings and found the PETS performance to be poor (1350 reward instead of 4600). This is because our CEM implementations differ in details; most significantly, we warm start our planner with the plans from the previous timestep, similar to [1]. This means our CEM yields significantly better results with a smaller number of planning trajectories. Given this difference in methods we feel the most fair comparison is to evaluate their method with the hyperparameters from their paper.
>
> The Clavera et al. paper was released less than 2 weeks before the ICLR deadline. Furthermore, it is a hybrid method (their actual policy is feedforward) and is thus not subject to the analysis performed in our paper. However, we will add a discussion of this work to our related work section.
>
> [1]: Tassa, Y., Erez, T., & Smart, W. D. (2008). Receding horizon differential dynamic programming. In Advances in neural information processing systems (pp. 1465-1472).

---

### Official Review · AnonReviewer4 · 2018-11-11
**More comprehensive comparison necessary**

**Rating:** 5
**Confidence:** 3

**Review:**

The paper proposes to use a multi-step prediction model in model-based RL. The proposed model maps from current state and a sequence of actions to the state after taking those actions. The paper demonstrates on 2 tasks that in a model-predictive control loop combined with planning by cross-entropy method, this can yield better asymptotic performance than using single-step models.

The insight of using multi-step prediction models is certainly appealing and makes a lot of sense in deterministic tasks. A systematic empirical comparison of multi-step deep models in RL is of interest, which this paper does provide to some extent.

An obvious limitation of the proposed deterministic multi-step forward model is the restriction to deterministic systems. One would expect that the performance deteriorates quickly as the system becomes more stochastic. An extension to the stochastic case along the lines of Chua et al, 2018 is non-trivial as capturing the stochasticity is typically more challenging in long-term predictions. Yet, the paper makes an additional assumption that is less clearly communicated: To be able to plan with a R-step model, one needs to be able to evaluate or approximate the sum of R rewards just from the first and last state in that R-long sequence. This work uses simply the reward at the end r(s_{t+R}) as a proxy which works well in these MuJoCo tasks but can fail horribly in others. One can imagine that a model not only outputs s_{t+R} but also the sum of R rewards given s_t and a_{t:t+R} which could work in more general settings but this is not explored in this paper. The contribution in this paper limited as the proposed approach as well as the experimental comparison is restricted to a relatively specific class of problems and no attempts to generalize are made.

The experiments nicely compare against using single-step dynamics models and the results show that using the multi-step models for MPC performs better in the two considered tasks. However, as fas as I understand both the ACP and Chua et al baseline using the single-step prediction accuracy to train their models. The paper is missing a comparison to single-step models that are trained using multi-step prediction losses ("backprop through time" as in Learning Nonlinear Dynamic Models by Langford et al 2009). These models should be much more robust to error blow-up for multi-step prediction and do not require the specific reward structure assumed in this paper.

The proposed R-step model-based RL approach could be connected to the use of options (the planner and model operate on R-step options, but the MPC does update the policy after every time step). It would be interesting to discuss this potential connection in the paper. The paper does a good job of discussing existing recent work in the deep RL literature but it would also be good to also discuss earlier work on multi-step prediction (e.g. in time-series modeling).

All in all, I think the paper makes a small contribution demonstrating that multi-step models are useful for model-based RL in specific domains -- which is interesting but certainly not surprising. Unfortunately the paper stops somewhat early by not comparing to relevant baselines (single-step models trained with multi-step losses) and by not considering tasks where the benefit of multi-step planning would be less clear.

---

> ### Author Response · Authors · 2018-11-26
> **Thank you for a comprehensive review!**
>
> We thank the reviewer for a very detailed review.
>
> "capturing the stochasticity is typically more challenging in long-term predictions": Yes, this is an interesting direction. Such a model would need the ability to produce complex multimodal distributions of outcomes. As our goal in this work was to understand why existing methods failed even on deterministic tasks, we leave it to future work to propose classes of multi-step models appropriate for stochastic environments.
>
> "This work uses simply the reward at the end": In fact, our model is able to use reward at intermediate time steps. To do this, we can use the variable-action-length version of our model, as used for visualization (mentioned in Section 2.2 paragraph 2 of the original version). We also use this version in Fig 5. to evaluate prediction error as a function of timesteps in the future. We have added Section 2.2.1 to clarify this.
>
> In our main experiments we use a transformed version of the reward function to select trajectories which have an outcome with the greatest x-axis position. This is approximately equal to the sum of the rewards obtained along the trajectory under the original reward function, which provides a reward at each timestep proportional to the forward progress at that timestep. We have updated Section 2.3.1 to better reflect this.
>
> We apologize for not making these points more clear in the paper.
>
> "single-step models that are trained using multi-step prediction losses": Yes, we agree that other models which directly minimize long-term prediction error will also produce better predictions that single-step models. However, the main goal of this work is to point out the deficiencies of the single-step class of models typically used in the field. As such, we simply chose one simple member of the class of multistep models to illustrate our point.
>
> We have since run experiments to evaluate the potential of recurrent training of single-step models with multi-step loss functions. While using a multi-step loss provides significant improvement over a single-step loss, the PCP remains somewhat more accurate. We have updated Figure 5 and the related text with this result.
>
> We furthermore have run these RNN models on the full bootstrapped planning task. Planning with these models is hugely slower than with the PCP, as for each prediction of length H the RNN must be run forward H times, whereas the PCP need only be run H/R times; that is, the RNNs take 50 times as long on Swimmer. We have included preliminary results with RNNs in Figure 7 and will update the paper with the completed experiments before camera ready. We do not believe these results alter the message of the paper.
>
> The Langford et al reference is very helpful and we will add a discussion of this paper to our related work.

---

### Author Response · Authors · 2018-11-26
**Understanding limits of existing models instead of proposing a new one**

We thank the reviewers for their constructive comments.

This work is primarily about understanding the limitations of existing neural network models for model-based RL, *not* introducing a new algorithm. In particular, we demonstrate why techniques like Nagabandi et al. and Chua et al. exhibit poor performance on some tasks which are superficially similar to the environments used in their papers.

Our paper clarifies the following issues in model-based RL:

- We show that MPC, even with a perfect model, needs long horizons to equal model-free performance on certain MuJoCo tasks. (We use Swimmer as a running example).
- We show that models of the standard form used by [Nagabandi, Chua] experience accumulating error with increasing prediction horizon, resulting in very poor long-term predictions.
- We verify that their poor reward performance is a consequence of their poor long-term accuracy, and is not e.g. intrinsic to neural network dynamics models or the interaction of such models with a planner. To do so we implement a simple modification to the form of the model which allows for good long-term prediction accuracy, which directly results in good task performance.
- We reveal the relationship between prediction accuracy and reward for MPC control.

---

### Meta-Review · Area_Chair1 · 2018-12-14
**Contribution narrower than current framing implies and should be enhanced and situated**

**Confidence:** 4
**Recommendation:** Reject

**Metareview:**

The issue of when model based methods can be used successfully in RL is an interesting one. However, the reviewers had a number of concerns about the significance and framing of this work with respect to the related literature. In addition, the abstract and title suggest a very generic contribution will be made, whereas the actual contribution is to a much more specific subclass. Some relevant papers (and their related efforts) include the following.

The Dependence of Effective Planning Horizon on Model Accuracy.
(AAMAS-15, best paper award) Nan Jiang, Alex Kulesza, Satinder Singh, Richard Lewis.

Self-Correcting Models for Model-Based Reinforcement Learning. Erik Talvitie. In 'Proceedings of the Thirty-First AAAI Conference on Artificial Intelligence (AAAI).' 2017.